# Chemical Composition and Phytotoxic, Antibacterial and Antibiofilm Activity of the Essential Oils of *Eucalyptus* *occidentalis*, *E. striaticalyx* and *E. stricklandii*

**DOI:** 10.3390/molecules27185820

**Published:** 2022-09-08

**Authors:** Marwa Khammassi, Flavio Polito, Ismail Amri, Sana Khedhri, Lamia Hamrouni, Filomena Nazzaro, Florinda Fratianni, Vincenzo De Feo

**Affiliations:** 1Nutritional Surveillance and Epidemiology Laboratory, National Institute of Nutrition and Food Technology, 11 rue Jebel Lakhdhar, Tunis 1007, Tunisia; 2Department of Pharmacy, University of Salerno, Via San Giovanni Paolo II, 132, 84084 Fisciano, Italy; 3Laboratory of Management and Valorization of Forest Resources, National Institute of Researches on Rural Engineering, Water and Forests, P.B. 10, Ariana 2080, Tunisia; 4Laboratory of Biotechnology and Nuclear Technology, National Center of Nuclear Science and Technology, Sidi Thabet, B.P. 72, Ariana 2020, Tunisia; 5Institute of Food Science, CNR-ISA, Via Roma, 64, 83100 Avellino, Italy

**Keywords:** *Eucalyptus*, essential oils, phytotoxic activity, antimicrobial activity, antibiofilm activity

## Abstract

The *Eucalyptus* genus (Myrtaceae) is characterized by a richness in essential oils (EO) with multiple biological activities. This study reports the chemical composition and the phytotoxic and antimicrobial activities of the EOs from Tunisian *E.*
*occidentalis*, *E.*
*striaticalyx* and *E.*
*stricklandii*. The EOs were analyzed using GC/MS and their phytotoxicities were assessed against the germination and seedling growth of *Sinapis arvensis*, *Trifolium campestre* and *Lolium rigidum*. Antimicrobial activity was investigated against both Gram-negative (*Pseudomonas aeruginosa, Escherichia coli* and *Acinetobacter baumannii)* and Gram-positive (*Staphylococcus aureus* and *Listeria monocytogenes*) bacteria. The inhibition of biofilm formation and its metabolism was determined at different times. All EOs were rich in oxygenated monoterpenes (36.3–84.8%); the EO of *E.*
*occidentalis* was rich in sesquiterpenes, both oxygenated and hydrocarbon (40.0% and 15.0%, respectively). Eucalyptol was the main constituent in all samples. The EOs showed phytotoxic activity on seed germination and seedling growth, depending both on chemical composition and weed. The EOs show a remarkable antibacterial potential resulting in a significant inhibition of the formation of bacterial biofilm and its metabolism, depending on the EO and the strain, with activity on the mature biofilm as well. Therefore, these *Eucalyptus* EOs could have potential applications both in the food and health fields.

## 1. Introduction

For a few years, the growing interest in natural products has been increased because of the ever-growing problem of antibiotic resistance and the use of synthetic products for pest control that have harmed both human health and the environment. Therefore, the exploration of natural substances from plants and trees appears as an alternative to the microbial resistance issue and a viable way find safe methods for weed control and management [1].

The genus *Eucalyptus*, which belongs to the Myrtaceae family, includes more than 700 species [2]. It is native to Australia and has become one of the most widely cultivated genera in the world [3]. In Tunisia, 117 Eucalyptus species have been introduced since 1957 [4], and scientific researchers, until today, continue to study their biological and pharmaceutical activities. *Eucalyptus* has been used in food and traditional medicine. The leaves of some *Eucalyptus* species have been proposed as a natural additive in the food industry and for health care [5], and their usefulness was also reported in the treatment of diseases and illness [6]. For example, leaves were used for respiratory infections, flu, and neurodegenerative and cardiovascular diseases [7,8].

Many species of this genus were recently reported for their broad spectrum of biological properties. *E. globulus* Labill., *E. citriodora* Hook. and *E. stricklandii* Maiden have been investigated for their antibacterial, cytotoxic, anti-inflammatory and antioxidant activities [9,10,11]. *E. occidentalis* Endl. has been studied for its insecticidal effects. Such activities can be attributed to the richness of eucalyptus in bioactive molecules, especially essential oils (EOs) that hold good promise as a replacement for antibiotics and herbicides [12]. EOs consist of a complex mixture of terpenes, phenols, ketones and aldehydes that act in synergy to bring the overall biological activities. Several reports investigated the antibacterial activities of *Eucalyptus* EOs [8] and demonstrated their action potential. *Eucalyptus* EOs can increase the permeability of the cell membranes of the bacterial strain, alter their microbial enzymes and cause cell death [8].

*Eucalyptus* EOs have also been investigated for their inhibitory potential on seed germination as well as their allelopathic activity [13]. They can inhibit the germination and growth of competing plants and have a spectrum of inhibitory effects against plants, pathogenic fungi and bacteria [14]. In addition, various bioactive molecules extracted from *Eucalyptus* species have allelopathic effects against weeds and microorganisms. However, according to our knowledge, no studies on the antibacterial and herbicidal activities of essential oils obtained from *Eucalyptus occidentalis* Endl., *E. striaticalyx* Maiden and *E. stricklandii* W. Fitzg. are known. Furthermore, their capacity to act against Gram-positive and Gram-negative bacteria biofilms, which let them increase their virulence and resistance to conventional antibiotics, is unknown.

We considered the current interest in *Eucalyptus* species and their multiple potentialities (especially antimicrobial and herbicidal activities). For this reason, our research aimed to study the chemical composition of EOs of three *Euc*al*yptus* species (*E. occidentalis*, *E. striaticalyx* and *E. stricklandii*). Furthermore, we evaluated their ability to inhibit biofilm formation and bacterial metabolism against Gram-negative (*Pseudomonas aeruginosa*, *Escherichia coli* and *Acinetobacter baumannii*) and Gram-positive bacteria (*Staphylococcus aureus* and *Listeria monocytogenes*). Moreover, this research reports data about the inhibitory effects of the three *Eucalyptus* EOs on seed germination and seedling growth of *Sinapis arvensis* L., *Trifolium campestre* Schreb. and *Lolium rigidum* Gaudin.

## 2. Results and Discussion

### 2.1. Essential Oil Yields

The essential oils were obtained by hydro-distillation of dried leaves in a Clevenger type apparatus, which were separated by the water phase. The extraction yield was calculated on a dry weight basis using the following formula:yield = (VEO × 100)/D.M
where D.M = dry material and VEO = volume of essential oil.

The average yield of the EOs varied according to the species, resulting in 1.62%*,* 2.51% and 1.83% (W/DW) for *E. occidentalis*, *E. striaticalyx* and *E. stricklandii*, respectively. The available literature reports yields of 1.1% and 2% respectively for EOs from leaves of Tunisian *E. occidentalis* and *E. stricklandii* harvested from Hajeb Layoun Arboreta (Kairouen, central Tunisia) [15]. Bignell and coworkers [16] reported a yield of 1.73% for the EO of *E. striaticalyx* from Australia.

Several researches found that the EO yield depends on the species and is mainly influenced by environmental and genetic factors [17,18]. In addition, the yield can be affected by the extraction conditions and the time of harvest, since temperature and season have an essential effect on the terpene biosynthesis, causing it to be higher in summer than in winter [19,20].

### 2.2. Chemical Composition

The EOs were analyzed using GC and GC-MS. Table 1 reports the composition of essential oils according to their elution order on a HP-5 MS capillary column. Figure 1 shows the main components of the analyzed essential oils.

Sixty-one components were identified, corresponding to 97.1% of the total EOs of *E. occidentalis* and *E. stricklandii* and to 96.5% of the total EO of *E. striaticalyx*.

*E. occidentalis* EO has significant proportions of oxygenated monoterpenes (41.0%) and oxygenated sesquiterpenes (40.0%). Eucalyptol (40.8%), viridiflorol (29.9%) and γ-patchoulene (5.5%) were the main components identified in this oil. Oxygenated monoterpenes were dominant in *E. stricklandii* EO, with an amount of 84.8%. Eucalyptol (73.6%), trans-pinocarveol (6%) and α-pinene (4.3%) were the main components.

*E. striaticalyx* EO was found to be richer in monoterpene hydrocarbons (43.5%) than in oxygenated monoterpenes (36.3%), with eucalyptol (29.6%), p-cymene (19.4%), spathulenol (8.7%), δ-2-carene (8.5%) and α-pinene (7.4%) as the main components. Eucalyptol was the most abundant compound in all analyzed EOs.

The EO compositions of *Eucalyptus* species from different countries have been investigated in several studies. The EO from Australian *E. occidentalis* leaves was rich in bicyclogermacrene (28.52%), α-pinene (18.85%), β-caryophyllene (5.44%), viridiflorol (5.13%) and globulol (4.31%) [21]. This composition is different from the Tunisian samples in the current work. However, another study conducted by Elaissi and coworkers [15] on *E. occidentalis* harvested from Hajeb Layoun Arboreta in Tunisia found that the leaf EOs were rich in oxygenated monoterpenes, with eucalyptol (18.8%), aromadendrene (13.2%), globulol (4.4%), pinene (6.9%) and trans-pinocarveol (5.6%) as the main components. Bande-Borujeni and coworkers (2018) reported that an EO of *E. occidentalis* leaves from Iran was rich in τ -cadinol (17.2%), 1,8-cineole (15.5%), α-cadinol (14%) and α-pinene (9.21%).

In agreement with our data, Bignell and coworkers [16] reported that EO from the leaves of *E. striaticalyx* from Australia was rich in 1,8 cineole (77.5%), *p*-cymene (4.8%) and aromadendrene (2.4%). Our results on the chemical composition of the EO of *E. striaticalyx* disagree with the composition of EO obtained from leaves harvested at Hajeb Laâyoun arboretum (Tunisia), rich in non-oxygenated monoterpenes with α-terpinene (25.7%), limonene (14.5%), α-pinene (8.5%) and α-thujene (3.6%) as the main components [19].

Our data on the EO from *E. stricklandii* leaves were similar in composition to those obtained by Elaissi and coworkers [15], but with different percentages. The EO from *E. stricklandii* leaves from Hajeb Layoun (Tunisia) was rich in oxygenated monoterpenes (34.3%) and monoterpene hydrocarbons (13.0%), with 1,8 cineole (20.4%), α-pinene (11.2%) and trans-pinocarveol (7.5%) as principal constituents.

In our study, *Eucalyptus* EOs were rich in oxygenated monoterpenes and eucalyptol was the principal constituent. However, we observed a particular variation in the chemical composition among species. This variation can be due to the differences in *Eucalyptus* species and their geographical origins [22]. The differences can also be attributed to ecological factors, such as climatic conditions, the state of plant material (dry or fresh), the origin of the plant, extraction methods and also the genetic diversity [18,22,23].

### 2.3. Phytotoxic Activity

The herbicidal activity of the EOs against three very aggressive weeds in Tunisia, *Sinapis arvensis* and *Trifolium campestre* (dicots) and *Lolium rigidum* (monocot), was evaluated (Figure 2). According to the results, remarkable phytotoxic effects were exerted by the tested EOs, resulting in an inhibition of germination and growth of the aerial and root parts of tested weeds, independently from the monocot or dicot class.

The data relating to germination inhibition effects are shown in Table 2. For all tested EOs, a dose–response effect was highlighted according to the statistical analysis.

For *E. occidentalis*, the inhibition of the germination of *S. arvensis* was total at the dose of 3 µL/mL, whereas, at the same dose, the inhibition was partial for *T. campestre* (20%) and *L*. *rigidum* (33%). These results testify the sensitivity of *S. arvensis* to *E. occidentalis* oils and the resistance of the other two seeds.

Similar results were noted for *E. stricklandii* and *E. striaticalyx* EOs. In fact, total inhibition of the germination of *S. arvensis* and *T. compestre* was recorded at 3µL/mL. However, at the same dose, the germination of *L. rigidum* was reduced to 6.7% and 10%with *E. stricklandii* and *E. striaticalyx* EOs, respectively.

In general, *L. rigidum* was the most resistant to the action of these EOs, while *S. arvensis* and *T. campestre* were the most sensitive. This behavior might be related to the selective resistance in mono and dicot weeds as described in recent papers [1,13]. In addition, *E. stricklandii* and *E. striaticalyx* EOs show a remarkable germination-inhibiting potential that exceeds the phytotoxic effect of *E. occidentalis*, which can be related to the different chemical compositions of the three tested species.

On the other hand, the action of the tested *Eucalyptus* EOs is accompanied by an inhibitory effect on the growth of the aerial parts when the inhibition of germination is partial (Table 3).

According to the statistical analysis, a dose–response effect was recorded. All three EOs show an apparent effect on the growth of the aerial parts of tested weeds, and *S. arvensis* was always the most sensitive.

In addition, the tested EOs exerted similar effects on the growth of the roots of all tested weeds. This inhibition result was dose dependent, depending both on the tested weed and the EO (Table 4).

This study first describes the phytotoxic effects of these EOs and agrees with the recent literature on EOs and crude extracts of *E. erythrocorys* F. Muell. that show a phytotoxic effect on the germination and growth of weeds, particularly against the germination and seedling growth of *S. arvensis* [13,24].

In a study carried out in Australia [25], the phytotoxic effects of 14 *Eucalyptus* EOs against the germination and growth of *Lolium rigidum* Gaudin (a monocot weed) were tested, resulting in a significant phytotoxic potential. Moreover, the authors evaluated the phytotoxicity of the EO components, with *trans*-pinocarveol and α-terpineol as the most active constituents.

In the current study, the EOs of *E. occidentalis*, *E. striaticalyx* and *E. stricklandii* were found to have variable chemical compositions, which can explain their different phytotoxic effects against the tested weeds. Several components are known for their herbicidal activity.

According to a study by De Martino and coworkers [26], 27 monoterpenes were tested for their anti-germinative activity against *Lepidium sativum* and *Raphanus sativus*. Some of those monoterpenes were also identified in the tested EOs.

The studied EOs show a specific richness in eucalyptol (1,8-cineole) (29.6–73.6%), an oxygenated monoterpene indeed known for its herbicidal effect. In fact, cineoles have been proposed as lead compounds for herbicides: cinmethylin, a commercial agrochemical, derives from 1,4-cineole (an isomer of eucalyptol), an oxygenated monoterpene found in EOs of various plants [27]. It inhibits germination and the growth of aerial and root parts; it also delays germination, inhibits chlorophyll pigment synthesis and inhibits cellular respiration [28]. Similarly, 1,8-cineole was reported to induce inhibition of germination and growth of weeds, and the application of these compounds induced a decrease in chlorophyll pigments and the mitotic index [29].

*p*-cymene, a compound representing 19.4% of the EO of *E. striaticalyx*, was known to possess herbicidal activity and completely inhibited the germination and seedling growth of *Amaranthus retroflexus* L., *Chenopodium album* L., and *Rumex crispus* L. [30]; this can explain the phytotoxic effects of *E. striaticalyx* EO. Sesquiterpenes are known for their allelopathic properties too, namely β-caryophyllene [31].

The mode of phytotoxic activity of EOs has been described in the literature. Their application generates oxidative stress, a release of malondialdehyde from the peroxidation of fatty acids of membrane phospholipids and an alteration of membrane integrity. The process induces a relative leakage of electrolytes and loss of vital membrane functions [32,33,34]. EOs disturb the synthesis of chlorophyll pigments, inducing an alteration of the energy balance [35]. In addition, Singh and coworkers [33] showed that the activities of antioxidant enzymes, such as superoxide dismutase, guaiacol peroxidase, catalase, ascorbate reductase and glutathione reductase, were significantly elevated, thereby indicating the enhanced generation of reactive oxygen species upon α-pinene exposure. Several monoterpenes affect chlorophyll content in plant seedlings; cell respiration; and the enzymatic activity of proteases, α- and β-amylases, peroxidases, and polyphenol oxidases in a dose-dependent way as a defense mechanism [34].

### 2.4. Antimicrobial Activity

Table 5 shows the minimal inhibitory concentration of the EOs needed to block the growth of the bacteria used as tester strains. Afterwards, we evaluated the ability of the EOs to affect bacterial adhesion and the mature bacterial biofilm. Then, we investigated the capacity of the EOs to act on the metabolism of the sessile cells, which can lead to an increase in bacterial virulence. These results are shown in Table 6 and Table 7.

Overall, the EOs were all able to act above all against *A. baumanni* and *S. aureus*, inhibiting, ab origine, the formation of biofilm. *A. baumanni* was very sensitive, and at the higher concentration, the inhibitory capacity of the EOs never dropped below 46.49% (*E. stricklandii* EO), reaching 86.73% (*E. occidentalis* EO). *S. aureus* was also sensitive to all EOs, which inhibited its adhesion capacity by up to 60.46%. This value was reached in the presence of the *E. striaticalyx* EO, which had a behavior similar to that of *A. baumannii*. When tested at the highest concentration, the three EOs were also active against *E. coli*, *L. monocytogenes* and *P. aeruginosa*, reaching inhibition rates of 79.48% (EO of *E. occidentalis* vs. *L. monocytogenes*). Only the EO of *E. stricklandii* was ineffective vs. *E. coli*, showing a poor ability to inhibit adhesion (1.5%).

The inhibitory efficacy on the adhesion capacity had similar results to the activity of the three EOs on the metabolism of bacterial cells: a 67.75% inhibition (*E. occidentalis* EO vs. *L. monocytogenes*) was registered and, in any case, was never lower than 17.43%. Exciting results were also provided by the test carried out on mature biofilms after 24 h of growth. In this case, the inhibitory action of the EOs was maintained but with different potency. Even when a decrease by 50% in the inhibitory activity of the EOs was registered, the EO of *E. stricklandii* kept its inhibitory force practically intact vs. *A. baumannii*. In other cases, the EOs proved to be much more effective on the mature biofilm than on the initial adhesion capacity of bacterial strains; thus, the EO of *E. stricklandii*, which vs. *L. monocytogenes* caused a 36.86% inhibition, proved to be extraordinarily active on the mature biofilm of this strain, resulting in an almost total inhibition (97.67%). The same applies to the EO of *E. striaticalyx*, which, already, at 10 μL/mL, almost totally inhibited the mature biofilm of *L. monocytogenes*. The EO of *E. stricklandii*, also able to inhibit the adhesion of *P. aeruginosa*, (42.66%), proved to work much more effectively on the mature biofilm of this bacteria, reaching 60.65% inhibition. In the case of *S. aureus*, all the EOs acted much more powerfully on the mature biofilm, reaching an inhibition up to 83.02% (10 μL/mL of *E. stricklanidii* EO).

From the metabolic point of view, it seems that the EOs worked mainly on the metabolism of sessile cells. However, the fact that the EOs were utterly incapable of acting on the metabolism of S. aureus may mean that their action could be different, working, for example, on the cellular structure or the genetic material [36].

Considering the chemical composition of the three EOs, it is possible to correlate their effectiveness in preventing or at least limiting the virulence of pathogenic strains to the presence, above all, of the eucalyptol. This compound is an active antimicrobial agent against many Gram-positive and Gram-negative pathogens, including those used in our experiments. It can act on the quorum sensing of bacteria, therefore upstream of the whole process that leads to the formation of the biofilm [37]. According to LaSarre and Federle [38], eucalyptol can act on the quorum sensing mechanism but not on the vital functions of *A. baumannii*. In the case of *S. aureus*, the presence of eucalyptol could have determined unwanted apoptosis, and, considering *E. coli*, the compound could have caused a robust condensation process of the nuclear chromatin present in its bacterial nucleosome [39].

Our results agree with Maczka and coworkers [39] as regards the effects of the EOs on the adhesion of *S. aureus*, but not about their action on the mature biofilm, where presumably the presence of other components may have somewhat held back the EOs’ effectiveness in acting on the metabolism of its sessile cells.

## 3. Materials and Methods

### 3.1. Plant Material

Leaves of *Eucalyptus occidentalis*, *E. striaticalyx* and *E. stricklandii* were collected at Djebbel Mansour arboretum of the governorate of Zaghouen, Tunisia. Six samples harvested from different trees equidistant at at least 20 m were collected for each species. The samples were then stored in a glass greenhouse for drying for 15 days. Mature seeds of annual weeds *Sinapis arvensis*, *Lolium rigidum* and *Trifolium campestre* were harvested from crop fields. The data about arboreta, climatic conditions, and the date of collection are listed in Table 8. Plants were identified by Professor Lamia Hamrouni, and voucher specimens were stored in the herbarium section of the Institut National de la Recherche en Génie Rural, Eaux et Forêts (INRGREF), Tunis.

### 3.2. Isolation and Analysis of the Essential Oils

The EOs were obtained by hydro-distillation of dried leaves in a Clevenger-type apparatus. The EOs were collected and dried over anhydrous sodium sulfate and stored in a brown glass bottle at 4 °C until used. Yield was calculated based on dried weight (*w*/*w*%).

GC and GC-MS were used to examine the composition of the essential oil. GC analyses were performed using a Perkin-Elmer Sigma 115 gas chromatograph equipped with a flame ionization detector (FID) and a non-polar HP-5 MS capillary column of fused silica (30 m × 0.25 mm; 0.25 μm film thickness). The operating conditions were the injector and detector temperatures of 250 °C and 290 °C, respectively. The analysis was conducted on a scheduled basis: 5 min isothermally at 40 °C; subsequently, the temperature was increased by two °C/min until 270 °C, and finally, it was kept in isotherm for 20 min. The analysis was also performed on an HP Innowax column (50 m × 0.20 nm; 0.25 μm film thickness). In both cases, helium was used as a carrier gas (1.0 mL/min). GC-MS analysis was performed using an Agilent 6850 Ser. II Apparatus equipped with a DB-5 fused silica capillary column (30 m × 0.25 mm; 0.25 μm film thickness) and connected to an Agilent Mass Selective Detector (MSD 5973); ionization voltage 70 V; ion multiplier energy 2000 V. The mass spectra were scanned in the range of 40–500 amu, with five scans per second. The chromatographic conditions were as reported above; transfer line temperature, 295 °C. Most of the components were identified by comparing their Kovats indices (Ki) with those in the literature [40,41,42] and by careful analysis of the mass spectra compared to those of pure compounds available in our laboratory or to those present in the NIST 02 and Wiley 257 mass libraries [43]. The Kovats indices were determined with a homologous series of n-alkanes (C10-C35) under the same operating conditions. For some compounds, the identification was confirmed by co-injection with standard compounds.

### 3.3. Phytotoxic Activity

The seeds of *Sinapis arvensis*, *Lolium rigidum* and *Trifolium campestre* were used in phytotoxic activity assays. Before germination tests, seeds were disinfected with 5% sodium hypochlorite, then rinsed with water. Twenty seeds were put in Petri dishes lined with double-layer filter paper Whatman No.1 and treated with different doses (0, 0.75, 1.50, 2.25 and 3.00 μL/mL) of *Eucalyptus* EOs in a solution of Tween 20 (0.1%) [13]. The tests were carried out completely randomized, with three replicates for each dose. After 12 days, the germination percentages were calculated, and roots and shoots growth were measured in cm. However, no significant differences (*p* ≤ 0.05) were found between negative control (pure water) and control (Tween solution), which explains the non-phytotoxic effects of Tween solution.

### 3.4. Antimicrobial Activity

#### 3.4.1. Microorganisms and Culture Conditions

*Acinetobacter baumannii* ATCC 19606, *Pseudomonas aeruginosa* DSM 50071 and *Escherichia coli* DSM 8579 (Gram-negative bacteria) and *Staphylococcus aureus* subsp. *aureus* Rosebach ATCC 25923 and *Listeria monocytogenes* ATCC 7644 (Gram-positive bacteria) were used as bacterial test strains. They were cultured in Luria broth for 18 h at 37 °C, and centrifuged at 80 rpm (Corning LSE, Pisa, Italy) before the microbial analysis. *A. baumannii* was cultured at 35 °C under the same conditions.

#### 3.4.2. Minimal Inhibitory Concentration (MIC)

The MIC of each EO was evaluated following the method described by Sarker and coworkers [44], modified as follows. The test was performed using 96 microtiter-plates. The resazurin solution was prepared by dissolving 270 mg homogenously in 40 mL of previously sterilized deionized water. 100 μL of samples, previously resuspended in 10% (*v*/*v*) DMSO, was put into the first row of the plate. To all other wells, we added 50 μL of Luria–Bertani broth or normal sterile solution. Then, we performed serial descending concentrations of our samples, and we added to each well 10 μL of resazurin indicator solution. 30 μL of 3.3× strength isosensitized broth and 10 μL of bacterial suspension (5 × 106 cfu/mL) were included in each well. The plates were well covered with parafilm to avoid dehydration, due to the little volume present in each well. As the positive control, wide-spectrum conventional antibiotic tetracycline (previously suspended in DMSO) was added in a column of the plate. Luria–Bertani broth was considered as the negative control. The plates were incubated at 37 °C (or at 35 °C for *A. baumannii*) for 24 h. The value of the MIC was revealed by the color change from dark purple to colorless. Tetracycline was chosen as a conventional antibiotic with a wide spectrum of action. This compound was used in other similar works [45,46].

### 3.5. Biofilm Inhibitory Activity

The capacity of the EOs to influence bacterial adhesion was investigated using flat-bottomed 96-well microtiter plates [47]. Before the test, the bacterial cultures were regulated to 0.5 McFarland with fresh culture broth. Then, 10 µL of the bacterial cultures and 10 or 20 µL/mL of the EOs were put in each well, and the wells were filled with different volumes of Luria–Bertani broth to have a final volume of 250 µL/well. The plates were coated with parafilm tape to avoid evaporation and incubated for 48 h at 37 °C (or at 35 °C for *A. baumannii*). Following the removal of the planktonic cells, sessile cells were delicately washed twice with sterile PBS, which was removed, and the plates were left for 10 min under a laminar flow hood. 200 µL of methanol was incorporated in each well to allow for the fixation of the sessile cells and discarded after 15 min. Each plate was left to allow for the dryness of the samples. The staining of the sessile cells was obtained by adding 200 µL of 2% *w*/*v* crystal violet solution/well. After 20 min, the staining solution was discarded; plates were softly washed with sterile PBS and left to dry. The release of the bound dye was allowed by adding 200 µL of glacial acetic acid 20% w/v. The absorbance was measured at λ = 540 nm (Cary Varian, Milano, Italy). The percent value of adhesion was calculated with respect to the control (formed by the cells grown without the presence of the samples, inhibition rate 0%). Triplicate tests were performed, and the average results were taken for reproducibility.

#### 3.5.1. Activity on Mature Bacterial Biofilm

The overnight bacterial cultures were adjusted to 0.5 McFarland with fresh Luria–Bertani culture broth, and 10 μL were added to flat-bottomed 96-well microtiter plates to have a final volume of 250 μL/well. Then, microplates were fully coated with parafilm tape to avoid evaporation and incubated at 37 °C (35 °C for *A. baumannii*). After 24 h of bacterial growth, the planktonic cells were removed, and the two concentrations of the EOs, 10 and 20 μL/mL, and Luria-Bertani broth were added to reach a final volume of 250 μL/well. After 24 h of incubation, the sequential steps of the experiment, including the calculation of the percent value of inhibition compared with the untreated bacteria, were performed as previously described.

#### 3.5.2. Effects of EOs on Cell Metabolic Activity within the Biofilm

The effect on the metabolic activity of the bacterial cells of two concentrations (10 and 20 μL/mL) of the EOs, which were added at the beginning of the bacterial growth and after 24 h of incubation, was also investigated through the 3-(4,5-dimethylthiazol-2-yl)-2,5-diphenyltetrazolium bromide (MTT) colorimetric method [46]. After 48 h total of incubation, the bacterial suspension, representing the planktonic cells, was removed; 150 μL of PBS and 30 μL of 0.3% MTT (Sigma, Milano, Italy) were added. The microplates were kept at 37 °C (35 °C for *A. baumannii*). After two hours, the MTT solution was removed, two washing steps were performed with 200 μL of sterile physiological solution, and 200 μL of dimethyl sulfoxide (DMSO) were added to allow the dissolution of the formazan crystals, measured after two hours at λ = 570 nm (Cary Varian, Milano, Italy).

### 3.6. Statistical Analysis

Data were subjected to one-way analysis of variance (ANOVA) using the SPSS 18.0 software package. Differences between means were tested through the Student–Newman–Keuls test, and values with *p* ≤ 0.05 were considered significantly different.

## 4. Conclusions

The data obtained confirmed the vast literature on the phytotoxic and allelopathic activity of *Eucalyptus* EOs. These were effective against the Gram-positive and Gram-negative bacteria who are generally more reluctant to undergo the action of conventional antibiotics. It is generally challenging to find a substance capable of decreasing bacterial virulence and limiting all the mechanisms leading to increased bacterial aggression, leading to more difficulty in eradicating the infections they trigger. Therefore, the activity exhibited by these *Eucalyptus* EOs against the pathogens used in our experiments could be considered of noticeable meaning, both for food and health purposes. In fact, in the last years, the extension of some infections was correlated to the expansion of the presence, in several environments (including foods, workplaces and hospital), of some bacteria, such as *P. aeruginosa*, *S. aureus*, *E. coli*, *L. monocytogenes* and *A. baumannii*, which developed robust evolutionary drug resistance due to their careless use, often in situations where their application was to be considered inefficient and inappropriate. Their higher drug resistance lets them form biofilms more quickly, causing a critical problem for food and health. Thus, the interest in natural alternatives to prevent biofilm formation increased the search for natural agents as alternatives to conventional sanitizers to control the biofilm’s development. Furthermore, the EOs were capable of acting not only at the beginning of the biofilm formation process but also on the mature biofilm, when the bacterial cells are more protected in the polymeric niches, becoming less sensitive to the action of the conventional drugs. This fact allows for the hypothesizing of their possible use in both the health and food fields.

## Figures and Tables

**Figure 1 molecules-27-05820-f001:**
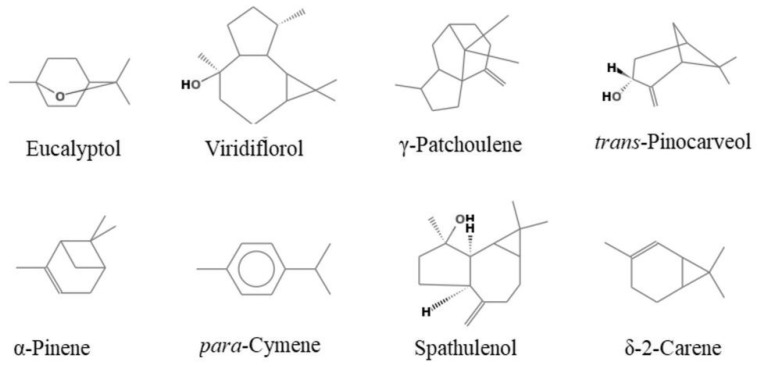
Main components of the analyzed essential oils.

**Figure 2 molecules-27-05820-f002:**
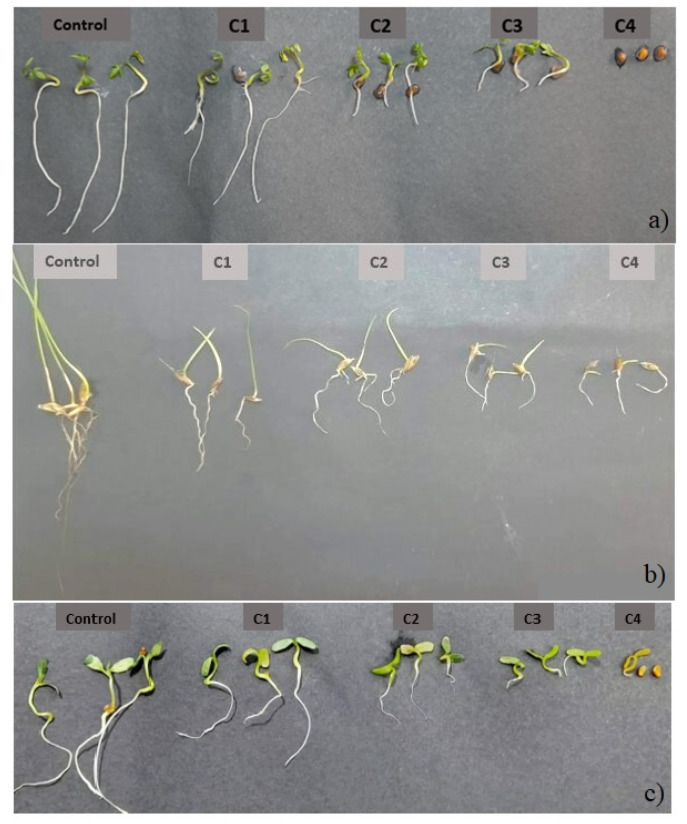
Phytotoxic effects of the studied EOs on *S. arvensis* (**a**), *L. rigidum* (**b**) and *T. campestre* (**c**).

**Table 1 molecules-27-05820-t001:** Chemical composition (%) of essential oils of *E. occidentalis* (A), *E.*
*striaticalyx* (B) and *E.*
*stricklandii (C)*.

	Compound	Ki ^a^	Ki ^b^	A%	B%	C%	Identification
1	α-thujene	930	-	-	2.1	-	1,2,3
2	α-pinene	939	1012	0.2	7.4	4.3	1,2,3
3	Camphene	954	1075	-	-	0.1	1,2,3
4	β-pinene	979	1110	-	1.8	-	1,2,3
5	Myrcene	990	1145	-	1.5	-	1,2,3
6	δ-2-carene	1002	-	-	8.5	-	1,2
7	α-phellandrene	1002	1177	0.1	-	-	1,2,3
8	α-terpinene	1017	1170	-	1.3	-	1,2,3
9	*p*-cymene	1024	-	0.8	19.4	0.6	1,2,3
10	Eucalyptol	1031	1031	40.8	29.6	73.6	1,2,3
11	γ-terpinene	1059	1221	-	0.6	1.0	1,2,3
12	*Cis*-sabinene hydrate	1070	-	-	0.1	-	1,2
13	Terpinolene	1088	-	-	0.9	-	1,2,3
14	*Exo*-fenchol	1121	-	-	-	0.2	1,2
15	*Cis-p-*menth-2-en-1-ol	1121	-	-	0.3	-	1,2
16	α-camphonenal	1126	-	-	-	0.1	1,2
17	*Trans*-pinocarveol	1139	-	0.2	0.3	6.0	1,2
18	Isopulegol	1149	1533	-	0.3	-	1,2
19	Eucarvone	1150	-	-	-	1.3	1,2
20	Sabina ketone	1159	1651	-	0.4	-	1,2
21	Isoborneol	1160	1642	-	-	0.4	1,2
22	Pinocarvone	1164	1586	-	0.1	-	1,2,3
23	*Neo-iso*-pulegol	1171	-	-	-	0.2	1,2
24	Terpinen-4-ol	1177	1590	-	3.4	0.7	1,2,3
25	Cryptone	1185	1659	-	0.4	0.2	1,2
26	α-terpineol	1188	1661	-	0.4	1.2	1,2,3
27	*Neo*-dihydrocarveol	1194	-	-	-	0.4	1,2
28	*Trans*-pulegol	1214	-	-	0.3	-	1,2
29	*Trans*-carveol	1216	1878	-	0.2	0.5	1,2
30	*Cis*-pulegol	1229	-	-	0.4	-	1,2
31	Cumin aldehyde	1241	-	-	0.1	-	1,2
32	δ-elemene	1338	1479	0.1	1.5	-	1,2
33	β-longipinene	1400	-	1.0	-	-	1,2
34	Longifolene	1407	1574	4.8	-	-	1,2
35	Aromadendrene	1441	1631	0.7	0.2	0.1	1,2
36	α-himachalene	1451	-	0.1	0.4	-	1,2
37	*Allo*-aromadendrene	1460	1660	0.1	-	0.1	1,2
38	9-*epi*-(E)-Caryophyllene	1466	-	0.1	0.1	-	1,2
39	γ-gurjunene	1477	-	0.6	0.6	-	1,2
40	*Epi*-cubebol	1494	-	0.1	0.1	-	1,2
41	Viridiflorene	1496	-	-	0.4	0.1	1,2
42	γ-patchoulene	1502	-	5.5	-	0.3	1,2
43	*Trans*-β-guaiene	1502	-	2.0	-	-	1,2
44	Spathulenol	1578	-	3.0	8.7	0.4	1,2
45	Globulol	1590	2104	-	2.1	1.8	1,2
46	Viridiflorol	1592	2110	29.9	-	-	1,2
47	Cubeban-11-ol	1595	-	1.0	0.2	-	1,2
48	Rosifoliol	1600	-	-	0.4	0.1	1,2
49	Guaiol	1600	2094	-	-	0.1	1,2
50	α-*epi*-7-*epi*-5-eudesmol	1607	-	2.0	-	-	1,2
51	*Epi*-cedrol	1619	-	2.1	-	-	1,2
52	γ-eudesmol	1632	2178	-	-	0.1	1,2
53	*Allo*-aromadendrene epoxide	1641	-	-	0.2	-	1,2
54	Cubenol	1646	2080	1.1	-	-	1,2
56	β-eudesmol	1650	-	-	0.2	3.2	1,2
57	Vulgarone B	1651	-	-	0.9	-	1,2
58	Cedr-8(15)-en-9-α-ol	1651	-	-	0.2	-	1,2
59	α-eudesmol	1653	2247	0.8	0.8	-	1,2
60	α-cadinol	1654	2224	-	0.3	-	1,2
61	5-hydroxy-isobornyl isobutanoate	1658	-	-	0.2	-	1,2
**Total**				**97.1**	**96.5**	**97.1**	
Monoterpene hydrocarbons				1.1	43.5	6	
Oxygenated monoterpenes				41.0	36.3	84.8	
Sesquiterpene hydrocarbons				15.0	3.2	0.6	
Oxygenated sesquiterpenes				40.0	13.5	5.7	

^a, b^: Kovats retention indices determined relative to a series of *n*-alkanes (C10–C35) on the apolar HP-5 MS and the polar HP Innowax capillary columns, respectively; c: identification method: 1 = comparison of the Kovats retention indices with published data, 2 = comparison of mass spectra with those listed in the NIST 02 and Wiley 275 libraries and with published data, and 3 = coinjection with authentic compounds; - = absent.

**Table 2 molecules-27-05820-t002:** Inhibitory effect of the EOs on percentage germination of *Lolium rigidum*, *Sinapis arvensis* and *Trifolium campestre*.

Weeds	Doses (µL/mL)	Essential Oils		
		*E. occidentalis*	*E. stricklandii*	*E. striaticalyx*
*L. rigidum*	Control (water)	86.66 ± 2.88 ^a^	86.66 ± 2.88 ^a^	86.66 ± 2.88 ^a^
	0	81.7 ± 1.7 ^a^	81.7 ± 1.7 ^a^	81.7 ± 1.7 ^a^
	0.75	78.3 ± 1.7 ^a^	61.7 ± 3.3 ^b^	70.0 ± 2.9 ^b^
	1.50	55.0 ± 2.9 ^b^	45.0 ± 5 ^c^	46.7 ± 4.4 ^c^
	2.25	33.3 ± 4.4 ^c^	31.7 ± 6 ^d^	26.6 ± 3.3 ^d^
	3.00	33.3 ± 1.6 ^c^	6.7 ± 1.7 ^e^	10.0 ± 2.9 ^e^
*S. arvensis*	Control (water)	96.66 ± 2.88 ^a^	96.66 ± 2.88 ^a^	96.66 ± 2.88 ^a^
	0	96.7 ± 1.7 ^a^	96.7 ± 1.7 ^a^	96.7 ± 1.7 ^a^
	0.75	70.0 ± 2.9 ^b^	63.3 ± 3.3 ^b^	66.7 ± 4.4 ^b^
	1.50	46.7 ± 4.4 ^c^	31.7 ± 1.7 ^c^	26.7 ± 4.4 ^c^
	2.25	23.3 ± 1.7 ^d^	16.7 ± 4.4 ^d^	6.7 ± 1.^7 d^
	3.00	1.7 ± 1.7 ^e^	0 ± 0 ^e^	0 ± 0 ^d^
*T. campestre*	Control (water)	93.66 ± 2.9 ^a^	93.66 ± 2.9 ^a^	93.66 ± 2.9 ^a^
	0	85 ± 2.9 ^a^	85.0 ± 2.9 ^a^	85.0 ± 2.9 ^a^
	0.75	75 ± 2.9 ^a^	63.3 ± 4.4 ^b^	46.7 ± 4.4 ^b^
	1.50	38.3 ± 4.4 ^b^	38.3 ± 1.7 ^c^	23.3 ± 1.7 ^c^
	2.25	35.0 ± 2.9 ^b^	21.7 ± 1.7 ^d^	8.3 ± 3.3 ^d^
	3.00	20.0 ± 2.9 ^c^	0 ± 0 ^e^	0 ± 0 ^d^

Results are reported as the mean ± SD of three experiments. The superscripts are used to compare the dose effect for each oil and on each weed. Means followed by the same letter are not significantly different by the Student–Newman–Keuls test (*p* ≤ 0.05).

**Table 3 molecules-27-05820-t003:** Inhibitory effect of the EOs on shoot growth (cm) of *Lolium rigidum*, *Sinapis arvensis* and *Trifolium campestre*.

Weeds	Doses (µL/mL)	Essential Oils		
		*E. occidentalis*	*E. stricklandii*	*E. striaticalyx*
*L. rigidum*	Control (water)	6.7 ± 0.5 ^a^	6.7 ± 0.5 ^a^	6.7 ± 0.5 ^a^
	0	6.4 ± 0.6 ^a^	6.4 ± 0.6 ^a^	6.4 ± 0.6 ^a^
	0.75	3.6 ± 0.3 ^b^	4.8 ± 0.21 ^b^	5.7 ± 0.3 ^ab^
	1.50	2.7 ± 0.3 ^bc^	4.4 ± 0.23 ^b^	4.7 ± 0.3 ^b^
	2.25	1.4 ± 0.2 ^c^	2.7 ± 0.2 _c_	3.4 ± 0.3 ^c^
	3.00	1.4 ± 0.1 ^c^	1.8 ± 0.3 ^c^	1.5 ± 0.2 ^d^
*S. arvensis*	Control (water)	8 ± 0.3 ^a^	8 ± 0.3 ^a^	8 ± 0.3 ^a^
	0	8 ± 0.46 ^a^	8.0 ± 0.46 ^a^	8 ± 0.46 ^a^
	0.75	5.9 ± 0.26 ^b^	5.0 ± 0.3 ^b^	6.7 ± 0.4 ^b^
	1.50	3.6 ± 0.23 ^c^	3.2 ± 0.1 ^c^	4.2 ± 0.2 ^c^
	2.25	2.1 ± 0.3 ^d^	2.3 ± 0.2 ^d^	1.9 ± 0.3 ^d^
	3.00	0.1 ± 0.1 ^e^	0 ± 0 ^e^	0 ± 0 ^e^
*T. campestre*	Control (water)	7 ± 0.4 ^a^	7 ± 0.4 ^a^	7 ± 0.4 ^a^
	0	6.9 ± 0.48 ^a^	6.9 ± 0.48 ^a^	6.9 ± 0.48 ^a^
	0.75	5.7 ± 0.2 ^b^	5.8 ± 0.4 ^b^	6.6 ± 0.3 ^a^
	1.50	4.3 ± 0.3 ^c^	3.9 ± 0.2 ^c^	4.3 ± 0.3 ^b^
	2.25	2.7 ± 0.2 ^d^	2.7 ± 0.4 ^d^	2.1 ± 0.2 ^c^
	3.00	1.3 ± 0.1 ^e^	0 ± 0 ^e^	0 ± 0 ^d^

Results are reported as the mean ± SD of three experiments. The superscripts are used to compare the dose effect for each oil and on each weed. Means followed by the same letter are not significantly different by the Student–Newman–Keuls test (*p* ≤ 0.05). Data are expressed in cm.

**Table 4 molecules-27-05820-t004:** Inhibitory effect of the EOs on root growth (cm) of *Lolium rigidum*, *Sinapis arvensis* and *Trifolium campestre*.

Weeds	Doses (µL/mL)	Essential Oils		
		*E. occidentalis*	*E. stricklandii*	*E. striaticalyx*
*L. rigidum*	Control (water)	5.5 ± 0.3 ^a^	5.5 ± 0.3 ^a^	5.5 ± 0.3 ^a^
	0	5.7 ± 0.3 ^a^	5.7 ± 0.3 ^a^	5.7 ± 0.3 ^a^
	0.75	5.3 ± 0.3 ^a^	4.9 ± 0.2 ^a^	3.7 ± 0.2 ^b^
	1.50	3.2 ± 0.2 ^b^	3.5 ± 0.3 ^b^	2.2 ± 0.1 ^c^
	2.25	1.3 ± 0.1 ^c^	2.3 ± 0.3 ^c^	2.2 ± 0.1 ^c^
	3.00	0.5 ± 0.2 ^d^	1.4 ± 0.1 ^d^	0.3 ± 0.1 ^d^
*S. arvensis*	Control (water)	6.9 ± 0.4 ^a^	6.9 ± 0.4 ^a^	6.9 ± 0.4 ^a^
	0	6.8 ± 0.32 ^a^	6.8 ± 0.32 ^a^	6.8 ± 0.32 ^a^
	0.75	6.5 ± 0.3 ^a^	4.1 ± 0.1 ^b^	5.0 ± 0.2 ^b^
	1.50	4.7 ± 0.4 ^b^	2.3 ± 0.2 ^c^	2.4 ± 0.2 ^c^
	2.25	2.9 ± 0.2 ^c^	1.5 ± 0.2 ^d^	2 ± 0.1 ^c^
	3.00	0.1 ± 0.1 ^d^	0 ± 0 ^e^	0 ± 0 ^d^
*T. campestre*	Control (water)	6 ± 0.2 ^a^	6 ± 0.2 ^a^	6 ± 0.2 ^a^
	0	5.6 ± 0.3 ^a^	5.6 ± 0.3 ^a^	5.6 ± 0.3 ^a^
	0.75	5.1 ± 0.2 ^a^	4.9 ± 0.1 ^b^	3.7 ± 0.5 ^b^
	1.50	3.2 ± 0.2 ^b^	2.7 ± 0.3 ^c^	1.8 ± 0.2 ^c^
	2.25	2.3 ± 0.4 ^c^	2.5 ± 0.2 ^c^	1.3 ± 0.1 ^c^
	3.00	1.4 ± 0.1 ^d^	0 ± 0 ^d^	0 ± 0 ^d^

Results are reported as the mean ± SD of three experiments. The superscripts are used to compare the dose effect for each oil and on each weed. Means followed by the same letter are not significantly different by the Student–Newman–Keuls test (*p* ≤ 0.05). Data are expressed in cm.

**Table 5 molecules-27-05820-t005:** MIC (µL/mL) of the EOs necessary to inhibit the growth of *A. baumannii*, *E. coli*, *L. monocytogenes, P. aeruginosa* and *S. aureus*. Tetracycline was used as positive control.

EO	*A. baumannii*	*E. coli*	*L. monocytogenes*	*P. aeruginosa*	*S. aureus*
*E. occidentalis*	25 ± 2 *	30 ± 3 *	35 ± 2	45 ± 2 ****	40 ± 2
*E. striaticalyx*	30 ± 3	40 ± 3 ****	40 ± 2 *	40 ± 3 *	35 ± 3
*E. stricklandii*	35 ± 3	40 ± 3 ****	35 ± 2	35 ± 3	35 ± 3
Tetracycline	31 ± 1	24 ± 3	33 ± 2	34 ± 2	38 ± 1

The experiments were performed in triplicate and reported as the mean ±SD. Results are reported as the mean ± SD of three experiments. *: *p* < 0.05, ** *p* < 0.01, ***: *p* < 0.001, ****: *p* < 0.00001 vs. control (tetracycline), according to two-way ANOVA followed by Dunnet’s multiple comparison test, at the significance level of *p* < 0.05.

**Table 6 molecules-27-05820-t006:** Percent inhibition of two doses of the EOs on biofilm formation of *A. baumannii*, *E. coli*, *L. monocytogenes, P. aeruginosa* and *S. aureus*, at 0 and 24 h.

Time 0	*A. baumannii*	*E. coli*	*L. monocytogenes*	*P. aeruginosa*	*S. aureus*
*E. occidentalis* 10 μL/mL	83.92 ± 2,17 ****	20.54 ± 0.07 ****	0 ± 0	0 ± 0	2.02 ± 0.74
*E. occidentalis* 20 μL/mL	86.73 ± 1.16 ****	46.93 ± 0.87 ****	79.48 ± 0.36 ****	13.79 ± 1.00 ****	31.08 ± 0.73 ****
*E. striaticalyx* 10 μL/mL	35.57 ± 6.98 ****	0 ± 0	8.23 ± 0.69 ****	0.84 ± 0.20	48.25 ± 2.12 ****
*E. striaticalyx* 20 μL/mL	66.66 ± 0.94 ****	10.36 ± 4.61 ****	27.16 ± 4.11 ****	30.67 ± 2.49 ****	60.46 ± 0.66 ****
*E. stricklandii* 10 μL/mL	5.24 ± 0.65 **	1.12 ± 0.40	0 ± 0	0 ± 0	45.20 ± 1.92 ****
*E. stricklandii* 20 μL/mL	46.49 ± 3.62 ****	1.48 ± 0.27	36.86 0.78 ****	42.66 ± 1.33 ****	45.16 ± 2.01 ****
24 h	*A. baumannii*	*E. coli*	*L. monocytogenes*	*P. aeruginosa*	*S. aureus*
*E. occidentalis* 10 μL/mL	9.71 ± 0.67 ****	28.60 ± 1.04 ****	0 ± 0	47.97 ± 0.58 ****	49.93 ± 0.82 ****
*E. occidentalis* 20 μL/mL	32.45 ± 0.62 ****	40.35 ± 0.87 ****	67.75 ± 0.46 ****	56.20 ± 0.57 ****	52.42 ± 0.85 ****
*E. striaticalyx* 10 μL/mL	39.11 ± 0.38 ****	6.35 ± 0.73 ****	26.38 ± 1.35 ****	8.18 ± 0.57 ****	19.40 ± 0.79 ****
*E. striaticalyx* 20 μL/mL	45.23 ± 0.91 ****	31.21 ± 1.21 ****	34.14 ± 0.62 ****	17.43 ± 0.58 ****	23.29 ± 1.15 ****
*E. stricklandii* 10 μL/mL	25.72 ± 0.83 ****	0 ± 0	0 ± 0	9.23 ± 1.37 ****	29.10 ± 0.89 ****
*E. stricklandii* 20 μL/mL	58.83 ± 0.38 ****	29.22 ± 0.81 ****	42.28 ± 0.62 ****	51.60 ± 0.37 ****	36.04 ± 0.89 ****

The percentage of inhibition was calculated using the formula: OD_C_-OD_S_/ODC*100, where OD_C_ is the OD of the untreated bacteria, and ODs is the OD of the bacteria treated with samples. Results are reported as the mean ± SD of three experiments. *: *p* < 0.05, **: *p* < 0.01, ***: *p* < 0.001, ****: *p* < 0.00001 vs. control (inhibition = 0), according to two-way ANOVA followed by Dunnet’s multiple comparison test, at the significance level of *p* < 0.05.

**Table 7 molecules-27-05820-t007:** Percent inhibition of two doses of the EOs on biofilm metabolic activity of *A. baumannii*, *E. coli*, *L. monocytogenes*, *P. aeruginosa* and *S. aureus*, at 0 and 24 h.

Time 0	*A. baumannii*	*E. coli*	*L. monocytogenes*	*P. aeruginosa*	*S. aureus*
*E. occidentalis* 10 μL/mL	41.05 ± 3.27 ****	0 ± 0	0 ± 0	0 ± 0	57.44 ± 4.98 ****
*E. occidentalis* 20 μL/mL	55.18 ± 1.30 ****	7.70 ± 0.80 ****	44.69 ± 0.40 ****	14.33 ± 0.20 ****	72.03 ± 2.30 ****
*E. striaticalyx* 10 μL/mL	0 ± 0	12.50 ± 0.57 ****	96.01 ± 1.11 ****	44.35 ± 0.85 ****	67.18 ± 4.25 ****
*E. striaticalyx* 20 μL/mL	27.19 ± 1.19 ****	27.11 ± 1.32 ****	96.81 ± 1.31 ****	39.38 ± 0.85 ****	76.36 ± 0.99 ****
*E. stricklandii* 10 μL/mL	0.00 ± 0.00	0.00 ± 0.00	20.90 ± 0.67 ****	0 ± 0	79.44 ± 0.78 ****
*E. stricklandii* 20 μL/mL	42.06 ± 1.30 ****	21.28 ± 1.87 ****	97.67 ± 1.81 ****	60.65 ± 1.87 ****	83.02 ± 1.04 ****
24 h	*A. baumannii*	*E. coli*	*L. monocytogenes*	*P. aeruginosa*	*S. aureus*
*E. occidentalis* 10 μL/mL	0 ± 0	10.32 ± 0.57 ****	0 ± 0	0 ± 0	18.43 ± 0.24 ****
*E. occidentalis* 20 μL/mL	88.39 ± 1.50 ****	27.08 ± 0.68 ****	0 ± 0	9.73 ± 0.47 ****	30.54 ± 0.63 ****
*E. striaticalyx* 10 μL/mL	62.22 ± 1.26 ****	23.15 ± 0.76 ****	26.28 0.15 ****	44.40 ± 0.69 ****	0 ± 0
*E. striaticalyx* 20 μL/mL	67.91 ± 0.72 ****	24.05 ± 0.92 ****	26.55 ± 2.44 ****	45.11 ± 0.84 ****	0.68 ± 0.02
*E. stricklandii* 10 μL/mL	6.26 ± 0.22 ****	0 ± 0	27.15 ± 0.80 ****	42.53 ± 0.35 ****	0 ± 0
*E. stricklandii* 20 μL/mL	67.14 ± 0.15 ****	13.29 ± 0.34 ****	29.77 ± 0.53 ****	45.31 ± 0.88 ****	0 ± 0

The percentage of inhibition was calculated using the formula: OD_C_-OD_S_/ODC*100, where OD_C_ is the OD of the untreated bacteria, and ODs is the OD of the bacteria treated with samples. Results are reported as the mean ± SD of three experiments. *: *p* < 0.05, **: *p* < 0.01, ***: *p* < 0.001, ****: *p* < 0.00001 vs. control (inhibition = 0), according to two-way ANOVA followed by Dunnet’s multiple comparison test, at the significance level of *p* < 0.05.

**Table 8 molecules-27-05820-t008:** Species, date and site of harvest of the plant material.

Species/Part Collected	Date of Harvest	Harvest Site	Specimen	Bioclimatic Stage
*E. occidentalis*/leaves	July 2021	Djebel Mansour, Zaghouan	EOC2143	Semi-arid
*E. striaticalyx*/leaves			ES2145	
*E. stricklandii*/leaves			ESI2145	
*L. rigidum/*seeds	June 2021	Sidi Ismail, Beja	LR2105	Subhumid with mild winter
*S. arvensis*/seeds			SA2103	
*T. campestre/*seeds			TC2101	

## Data Availability

Data are available from the authors.

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
