# Peer review of "Chemical Composition and Phytotoxic, Antibacterial and Antibiofilm Activity of the Essential Oils of Eucalyptus occidentalis, E. striaticalyx and E. stricklandii"

_molecules, 2022, doi:10.3390/molecules27185820_

Round 1
Reviewer 1 Report
The paper by Vincenzo De Feo and colleagues is focused on the analysis and evaluation of bioactivity of Essential Oils of E. occidentalis, E. striaticalyx and E. stricklandii. The topic is interesting, and the paper well presented. I report some suggestions that the authors may take into consideration to improve their manuscript, which can be accepted for publication after minor revision.
Abstract is very well written and clear and provides insights on the results. On the other hand, I find keywords to be very generic and may not help the reader enough in retrieving the paper.
Paragraph “2.1. Essential oil yields” is very brief and does not provide information on how extracts were obtained and yields calculated.
Table 1 is very clear but, since Molecules is a chemistry-oriented journal, a figure depicting the most relevant compounds could be useful to facilitate the reader.
Phytotoxic activity is well described, and results are clearly reported. Again, I strongly believe that the use of figures/photographs would really help in understanding the interpretation of results.
Antimicrobial activity: the authors may discuss why they decided to use tetracycline as positive control.
Author Response
Dear Referee, Thank you for your suggestion that will improve our manucript. The point-by-point modifications are reported in the enclosed file

Reviewer 2 Report
Comments to the Authors
The manuscript entitled “Chemical composition and phytotoxic, antibacterial and antibiofilm activity of the essential oils of E. occidentalis, E. striaticalyx, and E. stricklandii.” has been reviewed. The manuscript is interesting and complete, I suggest to accept the MS in the present form.
Author Response
The manuscript entitled “Chemical composition and phytotoxic, antibacterial and antibiofilm activity of the essential oils of E. occidentalis, E. striaticalyx, and E. stricklandii.” has been reviewed. The manuscript is interesting and complete, I suggest to accept the MS in the present form.
The Authors thank the Referee very much for its opinion.
Reviewer 3 Report
The manuscript authored by Khammassi et al. entitled “ Chemical composition and phytotoxic , antibacterial and Antibiofilm activity of the essential oils of E. Occidental’s, E. Striaticalyx, and E. Stricklandii” reported the chemical composition and multiple biological activities of essential oils from three Tunisian eucalyptus. The comments are below:
- Line 98-Line 99, “ the molecular formula and the subclass of each component are listed in Table 1”, but there has no molecular formula in Table 1.
- The superscripts a, b, c in Table 2, Table 3 and Table 4 are hard to understand. I suggest to explain them or delete them.
- The phytotoxic activity test (Table 2, Table 3 and Table 4) has no positive control nor negative control.
- I suggest to condense the discussion in “2.3 Phytotoxic activity” , it is too long.
- Please describe the protocol of MIC test in detail.
- Please provide the formula of inhibition for the results of Table 6 and Table 7. The significance level comparison vs. control (inhibition = 0 ) is meaningless.
- Line 386 and Line 391, change “two hundred uL “ to “ 200 uL”.
Author Response
Dear Referee, thank you for your opinion that will 'improve our manuscript. We modified the manuscript following all your suggestions. The point-by-point modifications are reported in the enclosed file

Round 2
Reviewer 3 Report
The English of the Manuscript has to be checked by an English mother language.